# Normalizing Flows For Out-of-Distribution Detection via Latent Density Estimation

## Abstract

Out-of-distribution (OOD) detection is a critical task for safe deployment of learning systems in the open world setting. In this work, we investigate the use of latent density estimation via normalizing flows for the OOD task and present a fully unsupervised approach with no requirement for exposure to OOD data, avoiding researcher bias in OOD sample selection. This is a fully post-hoc method which can be applied to any pretrained model, and involves training a lightweight auxiliary normalizing flow model to perform the out-of-distribution detection via density thresholding. Experiments on OOD detection in image classification show strong results for far-OOD data detection with only a single epoch of flow training, including 98.2% AUROC for ImageNet-1k vs. Textures, which exceeds the state of the art by 8.4%. Further, we provide insights into training pitfalls that have plagued normalizing flows for use in OOD detection.

## 1 Introduction

Machine learning has rapidly advanced in recent years, with state of the art models performing impressive tasks in a wide range of technical domains. However the standard workflow in machine learning is significantly less flexible than learning observed in animals in nature. While biological neural systems continually learn in uncontrolled environments, artificial neural networks are instead trained with a closed-world assumption (Yang et al., 2021) on a fixed corpus of training data, validated against a set of reserved data drawn from the same data distribution, and then deployed to perform roughly the same task. When deployed these models can be exposed to inputs that are dissimilar to the in-distribution (ID) data they were trained and validated on, and potentially leading to unpredictable behavior when encountering this out-of-distribution (OOD) data.

Addressing how artificial neural networks can be used in open world situations where they may be exposed to out-of-distribution data remains a challenge. Out-of-distribution detection is the task of identifying when a sample is not drawn from the training data distribution. This is especially important in safety critical applications such as autonomous vehicles; the statistical assurances on model performance provided by the validation dataset are no longer applicable.

In this work, we revisit using latent density estimation via normalizing flows for out-of-distribution detection in image classification. Prior works assert that normalizing flows are not effective for OOD detection when performing density estimation in pixel space (Nalisnick et al., 2019), and density estimation in the latent space of pretrained models has been discussed but not thoroughly investigated (Kirichenko et al., 2020). Contrary to works that dismiss normalizing flows in this domain, we demonstrate that by performing density estimation in the latent space of a pretrained image classification backbone model, normalizing the latent representations, and undertraining the normalizing flow we are able to achieve competitive results on both small and large datasets. The proposed method has the advantages of being fully unsupervised and requires no exposure to OOD training data, avoiding researcher bias from a specific definition of the OOD data. Finally, this is a post-hoc method that can be applied to any pretrained classification model, and it requires training a lightweight normalizing flow model for only a single epoch to perform the latent-space density estimation for out-of-distribution detection, making it a broadly applicable technique.

## 2 RELATED WORK

### 2.1 OUT-OF-DISTRIBUTION DETECTION

OOD detection deals with identifying semantically distinct samples (from unseen classes) to avoid erroneously classifying them as one of the classes in the training distribution. Out-of-distribution detection performance is evaluated by attempting to discriminate between a validation dataset versus an out-of-distribution dataset. The most widely used metric is the area under the receiver operating characteristic (AUROC) (Fawcett, 2006), a threshold-free classification performance metric useful for comparing unbalanced datasets. The false positive rate at 95% true positive rate (FPR95) is also employed to a lesser extent, as it provides a single point snapshot of false positive rate with a fixed OOD detection performance requirement.

OOD detection is a rich field with many existing approaches (Yang et al., 2021). These can be divided into classification-based, distance-based, generative-based, and density-based approaches. Classification-based approaches define a classification output that identifies ID and OOD inputs at inference time, with common baseline methods including the max-softmax probability (MSP, Hendrycks & Gimpel (2017)), ODIN (Liang et al., 2018), the energy score (Liu et al., 2020), and post-hoc methods that modify the penultimate layer activations such as ASH (Djurisic et al., 2023) and ReAct (Sun et al., 2021). MSP is a simple baseline method which thresholds on the maximum class probability. Energy score is a more modern development with stronger performance while remaining simple to implement, calculating a metric inspired by thermodynamics (the free energy) from the classification logits. ReAct is used in conjunction with the energy score, but clips the top 10% of latent variables off prior to evaluation, resulting in state of the art performance on large scale datasets. Distance-based methods label OOD samples as those sufficiently far from ID training sample feature vectors, and include Euclidean and Mahalanobis distance (Lee et al., 2018). Generative-based approaches employ generative models to reconstruct inputs, and assess samples with poor reconstruction accuracy or low likelihood under the generative model as OOD (Salehi et al., 2022). Examples of this approach include VAEs with modified priors (Floto et al., 2023), hierarchical VAEs (Havtorn et al., 2021), and diffusion models (Graham et al., 2023; Liu et al., 2023). These methods commonly require training a generative network to model the data distribution, which can be computationally demanding.

Finally, for density-based approaches a density estimation model is built from the training data such that the ID data lies within high density regions, and OOD data encountered at inference time occupies low density regions. A threshold on the density can be added to transform a density estimator into an out-of-distribution detector, classifying low probability data as out-of-distribution. In this approach the density estimator is used as a proxy for model epistemic uncertainty (Hüllermeier & Waegeman, 2021). Density estimation methods can be performed in the input data space or a transformed representation space, and include kernel methods, radial basis functions, and normalizing flows (Hüllermeier & Waegeman, 2021; Yang et al., 2021). Kirichenko et al. (2020) observed that using a normalizing flow to perform density estimation in the latent space improves over performing density estimation in the pixel space, but their analysis is extremely limited and their results do not exceed other state-of-the-art methods.

### 2.2 NORMALIZING FLOWS

Normalizing flows are a class of generative neural networks that are dimensionality preserving and fully invertible. They are trained to map between two probability distributions, typically a data distribution and a known base probability distribution, such as the normal distribution. Normalizing flows have the dual function of being an exact density estimator (by measuring the probability of a datapoint when mapped to the base distribution), and a generative model (by sampling from the base distribution, and then mapping into the data space). Mathematically, normalizing flows can be written as implementing a change of variables:

$$p(z) = q(f_\theta(z)) \left| \det \left( \frac{\partial f_\theta(z)}{\partial z^T} \right) \right|$$

$$\log p(z) = \log q(f_\theta(z)) + \log |\det (J f_\theta(z))|$$

where $p(z)$ is the data distribution, $q(z)$ is the known base distribution, and $f_\theta(z)$ is the mapping function between these two distributions, implemented as an invertible normalizing flow network parameterized by $\theta$.

For a more thorough and formal review of the mathematics of normalizing flows, we refer readers to Kobyzev et al. (2021). Implementing normalizing flows is often more challenging than other neural networks, as the model must be entirely invertible and should have a Jacobian that can be efficiently calculated. However, they have shown impressive performance in many tasks, including generating realistic images of faces (Kingma & Dhariwal, 2018) and high quality density estimation on image data (Ho et al., 2019).

### 2.3 NORMALIZING FLOWS FOR OUT-OF-DISTRIBUTION DETECTION

Normalizing flows have been applied to the task of out-of-distribution detection in several prior works with mixed success, but have historically performed very poorly for OOD detection in the image classification domain. When performing density estimation on pixel data in images, previous authors recommend against the use of normalizing flows, finding that they learn spurious pixel-level correlations and capture low-level statistics rather than high-level semantics (Nalisnick et al., 2019; Kirichenko et al., 2020; Zhang et al., 2021a).

In Gudovskiy et al. (2022), Rudolph et al. (2021), and Rudolph et al. (2022) normalizing flows are applied to image segmentation anomaly detection by performing density estimation of multiscale feature map embeddings instead of pixel space. Results are promising, but limited to small scale datasets, and they use hand-tailored network architectures that do not generalize to other domains.

Flows have also been used for anomaly detection in video data. Cho et al. (2022) uses a Glow normalizing flow (Kingma & Dhariwal, 2018) to perform density estimation of the latent variables produced by two autoencoders, one capturing spatial information and one capturing temporal information. This work highlights the importance of performing density estimation in the latent space and demonstrates competitive performance in this domain, but has a complex autoencoder architecture with a reconstruction loss term, limiting its potential applications. Jiang et al. (2022) apply a normalizing flow for the task of quantifying sample rareness, illustrating the value of latent space density estimation with normalizing flows for the downstream task of data mining and dataset balancing.

Zhang et al. (2020) demonstrate strong OOD detection performance using normalizing flows in image classification, but is not a post-hoc method, as it requires jointly training both the classifier backbone and the normalizing flow model with additional loss hyperparameters. This approach is limited by the necessity to jointly learn the latent space, and results are only evaluated on small datasets. Kirichenko et al. (2020) briefly introduce the concept of performing density estimation in the latent space of a pretrained classifier, but their analysis is very limited and results are not compelling. Our work carries the investigation of latent density estimation via normalizing flows investigation much further, demonstrating that normalizing flows can achieve state of the art out-of-distribution detection in image classification as a simple, post-hoc method with no complex architecture changes or modifications to the backbone.

## 3 METHOD

### 3.1 LATENT DENSITY ESTIMATION

In this work we leverage a pre-trained neural network backbone to provide a compressed, reduced representation of our input data that is rich in semantic information for the downstream task of image classification. We use the penultimate layer's activations as latent variables for density estimation. These latent variables contain all of the necessary information for the backbone model to perform the output classification task, and are typically transformed to the final output logits using a linear projection head.

We perform density estimation on the latent representations using existing normalizing flows (Kingma & Dhariwal, 2018; Dinh et al., 2017; Chen et al., 2019), learning an invertible mapping between the latent space and a normal probability distribution. Our normalizing flows are trained on the penultimate layer activations of a frozen pre-trained image classifier, with the optimization

criterion of minimizing the log likelihood of the transformed latents. As an unsupervised method, the class labels of the original image data are unused. Once trained, the normalizing flow is a computationally efficient density estimator for the latent space of the pre-trained backbone model. Out-of-distribution detection is achieved by applying a simple probability threshold to the density estimates for new samples, classifying low density samples as OOD. This straightforward technique is fully post-hoc, and only requires training a normalizing flow on ID data with no exposure to any OOD data. See Figures 1 and 2 below for a block diagram and visualization of our method.

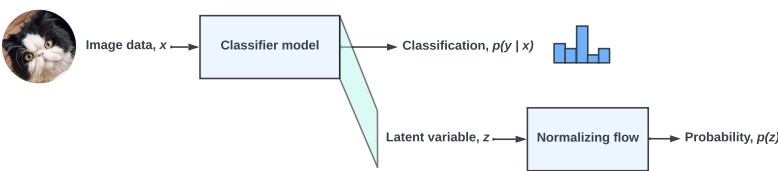

Figure 1: Visualization of the method of latent density estimation for out-of-distribution detection.

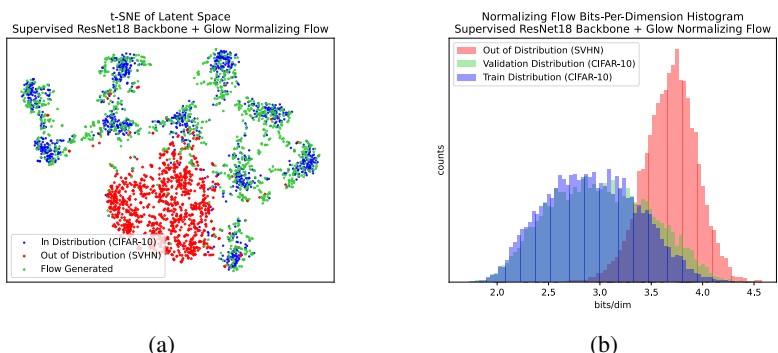

| (a) | (b) |

Figure 2: (a) A t-SNE visualization of the latent space of a ResNet18 backbone model comparing ID data (CIFAR-10), OOD data (SVHN), and flow generated latent variables. Clearly visible are 10 clusters corresponding to the 10 classes of CIFAR-10, as is the coincidence between the ID data and the flow generated data points. (b) A histogram of the bits-per-dimension to represent ID latents versus OOD latents under the normalizing flow model. Better separability of these distributions leads to higher AUROC for OOD detection.

### 3.2 MEASURING BITS-PER-DIMENSION

A connection has been established in the literature between out-of-distribution detection and lossless compression, and lossless compression and flow models (Zhang et al., 2021b; Yang et al., 2023). Since normalizing flows provide exact likelihood estimates, they can be used as the basis for a lossless compression scheme via Shannon's source coding theorem (Shannon, 1948). A dequantization step is common practice for density estimation of discrete data; random noise drawn from the uniform distribution $U(0, \frac{1}{256})^d$ is commonly used when modelling $d$-dimensional pixel data (Dinh et al., 2017; Lippe & Gavves, 2021; Theis et al., 2016), as each pixel can take on 256 values. We select a quantization precision of $\epsilon = \frac{1}{1024}$ as this is the smallest precision which half-precision floating-point, a common data format for neural networks, can represent around 1 (IEEE, 2008). When training flow models, latent variables are additively augmented with uniform random noise drawn from $U(-\frac{\epsilon}{2}, \frac{\epsilon}{2})^d$. We estimate the lower bound on the bits per dimension ($bpd$) needed to represent latent variable $z$ to a precision of $\epsilon = \frac{1}{1024}$ under an entropy coding scheme (Shannon, 1948) as:

$$bpd = -\frac{\log_2(p(z))}{d} - \log_2(\epsilon)$$

where $z \in \mathbb{R}^d$, and $p(z)$ is the probability of $z$ under our normalizing flow model. In-distribution data produces latent variables with a high probability under the flow model, which can be recon-

structed using a small amount of information under a lossless compression scheme. Conversely, out-of-distribution latent representations are assigned lower probability, requiring a larger amount of information to represent.

## 4    EXPERIMENTAL SETUP

We evaluate the utility of normalizing flows for out-of-distribution detection on a range of image classification tasks, using a variety of backbone networks and in-distribution datasets.

**Datasets**: We use CIFAR-10 (Krizhevsky, 2012) and ImageNet-1k (Deng et al., 2009) as our in-distribution datasets. CIFAR-10 models are evaluated against random Gaussian noise, SVHN (Netzer et al., 2011), Places365 (Zhou et al., 2018), CelebA (Liu et al., 2015), and CIFAR-100 (Krizhevsky, 2012) as out-of-distribution datasets. ImageNet-1k models are evaluated against Textures (Cimpoi et al., 2014) and reduced versions of iNaturalist (Van Horn et al., 2018), SUN (Xiao et al., 2010), and Places (Zhou et al., 2018). The latter three are filtered to ensure these datasets contain no common classes with ImageNet-1k, as done by Sun et al. (2021). Datasets can be considered as either *near-OOD* or *far-OOD* (Salehi et al., 2022) depending on how semantically distinct they are from the ID dataset; in this work we consider Gaussian noise and SVHN to be far-OOD vs. CIFAR-10, and Textures to be far-OOD from ImageNet-1k.

**Evaluation**: Out-of-distribution detection performance is evaluated using AUROC, calculated between the in-distribution validation dataset and out-of-distribution dataset. AUROC is a threshold-free metric, and an AUROC of 50% indicates no separability between the distributions, while an AUROC of 100% indicates perfect separability between the distributions. We evaluate our method against MSP (Hendrycks & Gimpel, 2017), ODIN (Liang et al., 2018), energy score (Liu et al., 2020), and ReAct (Sun et al., 2021).

**Normalizing Flow Models**: We use a 10 block Glow (Kingma & Dhariwal, 2018) flow for all experiments. For flow models trained on CIFAR-10, each block is composed of two linear layers with dimension [512, 2048, 512]. For flow models trained on ImageNet-1k, each block is composed of two linear layers which do not alter the dimensionality of the latent space (2048 for ResNet50 and 768 for Swin-S). Flow models are trained using the Adam optimizer (Kingma & Ba, 2014) for only a single epoch with a learning rate of 1e-4 for CIFAR-10 and 1e-5 for ImageNet-1k.

**Backbone Model**: For CIFAR-10, we train a ResNet18 classifier backbone using supervised learning to a validation accuracy of 92.1%. For ImageNet-1k, we use two Pytorch pretrained models as classifier backbones: ResNet50 and Swin-S (top-1 validation accuracies of 80.6% and 83.2% respectively).

## 5    RESULTS

We first present our main results for CIFAR-10, summarized Table 1. Our method exceeds other approaches on SVHN and Gaussian noise (far-OOD), and is competitive with alternatives on more challenging datasets (Places365, CelebA, CIFAR-100). Further, we present results for the larger scale ImageNet-1k dataset in Table 2. With a ResNet backbone, our method is able to achieve 98.2% AUROC on Textures (Cimpoi et al., 2014), and obtain 8.4% better performance than the next best method, ReAct (Sun et al., 2021). With a transformer backbone, we again outperform ReAct by 4.1% on Textures. Samples from the Textures dataset are visually distinct from ImageNet and can be considered far-OOD. Our method outperforms competing methods at detecting these more visually distinct types of OOD samples (CFIAR-10 vs. SVHN, and ImageNet vs. Textures).

Table 1: **Main results**, CIFAR-10.

| Backbone | Method | Out-of-Distribution Dataset (AUROC ↑) | | | | |
|---|---|---|---|---|---|---|
| | | **Gaussian** | **SVHN** | **Places365** | **CelebA** | **CIFAR-100** |
| ResNet18 + CIFAR-10 | MSP | 90.86 | 88.62 | 88.78 | 90.62 | 85.86 |
| | Energy Score | 87.33 | 87.12 | 92.77 | 94.62 | **88.35** |
| | ODIN | 88.95 | 87.47 | **95.13** | **95.16** | 87.43 |
| | LDE (ours) | **99.41** | **96.06** | 92.61 | 93.23 | 85.93 |

Table 2: **Main results**, ImageNet-1k. *Results from Sun et al. (2021).

| Backbone | Method | Out-of-Distribution Dataset (AUROC ↑) | | | |
|---|---|---|---|---|---|
| | | **Textures** | **iNaturalist** | **SUN** | **Places** |
| ResNet50 + ImageNet-1k | MSP | 77.75 | 86.15 | 81.48 | 79.77 |
| | ReAct* | 89.80 | **96.22** | **94.20** | **91.58** |
| | Energy Score | 84.08 | 87.92 | 87.35 | 83.97 |
| | ODIN | 85.45 | 88.82 | 87.13 | 83.73 |
| | LDE (ours) | **98.19** | 80.40 | 81.26 | 72.63 |
| Swin-S + ImagetNet-1k | MSP | 80.42 | 88.45 | 82.98 | 81.53 |
| | ReAct | 84.89 | 92.02 | **86.78** | **85.09** |
| | Energy Score | 77.97 | 81.40 | 75.47 | 72.13 |
| | ODIN | 63.10 | 63.96 | 56.44 | 50.8 |
| | LDE (ours) | **88.97** | **94.57** | 85.64 | 82.89 |

## 6 DISCUSSION

Normalizing flows are powerful tools for density estimation of high dimensional data, but their utility for out-of-distribution detection is contingent on the details of their training. To practically implement normalizing flows for OOD detection, we discuss several key considerations.

### 6.1 DIMENSIONALITY REDUCTION

The dimensionality of the latent representation plays a large role in the performance of the normalizing flow. Very high dimensional representations are more challenging to model and may require more data to train. In previous work, performing preliminary dimensionality reduction of the latent variables via principal component analysis (PCA) prior to density estimation has been used to help with computational efficiency in flow training (Jiang et al., 2022).

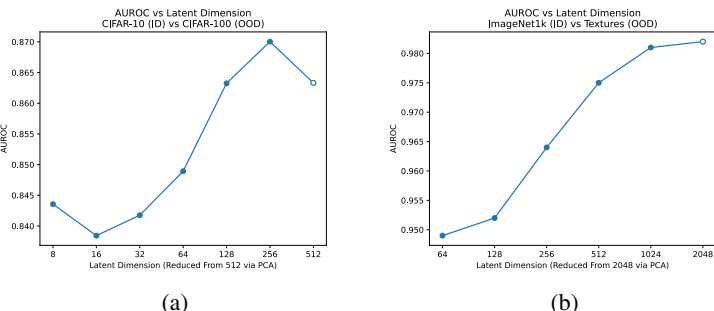

(a)         (b)

Figure 3: Impact of PCA dimensionality reduction of latent variables on OOD detection.

However, our experiments show that generally an unreduced latent space yields stronger OOD detection performance (Figure 3). Dimensionality reduction will inevitably remove information from the latent representations, and in contrast to labelled tasks or methods that use OOD data exposure, it is unclear what information in the latent representations is required for high performance OOD detection, and how to preserve this information. For the larger dataset, performance increases monotonically with latent dimension, while for CIFAR-10, the performance generally increases but is lower for the unreduced embedding of size 512. This is likely due to the increased training difficulty.

### 6.2 FLOW REGULARIZATION

During training, the goal is to optimize a flow model that generalizes from the training to the validation distribution, while still separating OOD data. It is critically important to manage overfitting: the normalizing flow's generalization gap between the training and validation data distributions directly

impacts the separability of validation and out-of-distribution data (see Figure 4 for a visualization of training, validation, and OOD bit-per-dimension distributions).

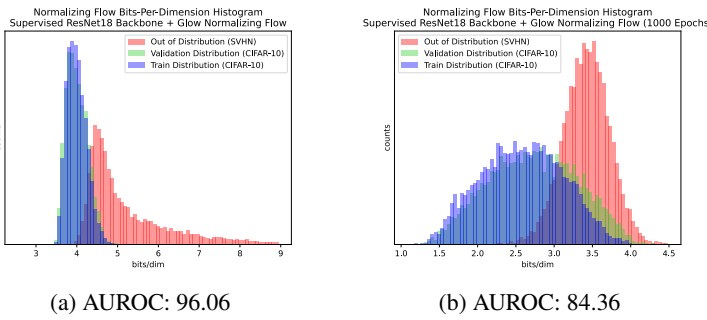

| (a) AUROC: 96.06 | (b) AUROC: 84.36 |

Figure 4: Bits-per-dimension histograms for the same flow model at 0 epochs and 1000 epochs. With further training, the bits/dim decreases for all distributions, but the training and validation distributions begin to separate due to overfitting, and the separability of the ID/OOD distributions degrades.

A variety of standard regularizations techniques can be used to avoid overfitting on the training data: augmenting the dataset, applying dropout, and carefully selecting the model size with respect to the amount of data available. We train our flows on latent representations obtained with dataset augmentation identical to the augmentations used to train the backbone model.

Under-training was found to be critically important. Our experiments demonstrated a surprising trend: training a normalizing flow model to minimize the validation loss may actually be detrimental to OOD detection performance. Loss on a test OOD dataset *decreases* during training (OOD data becomes more likely as the flow model fits to ID data), and AUROC for this test OOD dataset peaks early, then drops with additional epochs as the flow model fits to the training data (see Figure 5). The optimal number of epochs to train a flow model for depends on the OOD dataset, flow architecture, and backbone model, and is far before the validation loss begins to rise (classic overfitting). This is thus distinct from early stopping, and represents a novel and beneficial form of under-training. In our work we report all results using normalizing flows trained to only a single epoch to ensure consistent evaluations across models and datasets.

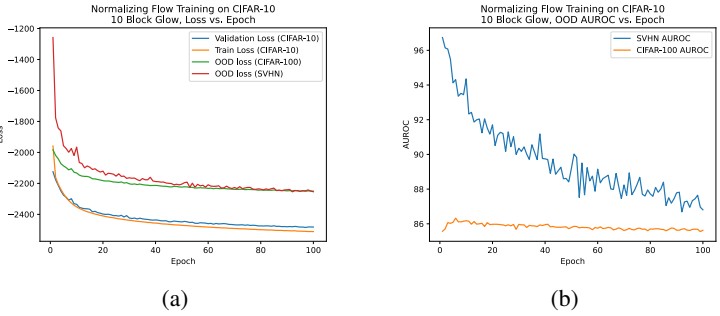

| (a) | (b) |

Figure 5: Normalizing flow loss and AUROC during training. Loss evaluated on both ID and OOD test datasets decreases during training, and AUROC peaks early and then declines.

### 6.3 SCALING LATENT REPRESENTATIONS

We find that normalizing the latent variables strongly improves OOD detection performance on far-OOD datasets (CIFAR-10 vs. SVHN and ImageNet-1k vs. Textures, for example). We can write the final linear head of the backbone model as:

$$y = W^T z = \|z\|(W^T \hat{z})$$

Where $W$ is the weight matrix (ignoring the bias term), $z$ is our latent variable, $\|z\|$ is the Euclidean norm of $z$, and $\hat{z}$ is the normalized (unit-length) latent variable. We interpret the product $W^T \hat{z}$ as

the semantic agreement between $\hat{z}$ and each logit's class, while $\|z\|$ relates to the network's overall confidence in the output. By training the normalizing flow model on normalized latent variables, density estimation is performed on the semantic content of the latent space disentangled from the network's confidence. As a result, experiments show that performance is substantially improved on far-OOD data.

Table 3: Flow AUROC results, CIFAR-10: normalized vs. unnormalized latents

| Backbone Model | Latent Scaling | Out-of-Distribution Dataset (AUROC ↑) | |
| --- | --- | --- | --- |
| | | SVHN | CIFAR-100 |
| ResNet18 + CIFAR-10 | Unnormalized | 85.63 | 57.68 |
| | Normalized | **96.48** | **85.93** |

Table 4: Flow AUROC results, ImageNet-1k: normalized vs. unnormalized latents

| Backbone Model | Latent Scaling | Out-of-Distribution Dataset (AUROC ↑) | | | |
| --- | --- | --- | --- | --- | --- |
| | | Textures | iNaturalist | SUN | Places |
| ResNet50 + ImageNet-1k | Unnormalized | 86.95 | 49.78 | 60.21 | 58.89 |
| | Normalized | **98.19** | **80.40** | **81.26** | **72.63** |
| Swin-S + ImageNet-1k | Unnormalized | 86.57 | 93.70 | **86.63** | **85.10** |
| | Normalized | **88.97** | **94.57** | 85.64 | 82.89 |

## 6.4 FLOW ARCHITECTURE

Normalizing flow architecture is an active area of research, with different flow designs having their own pros and cons. RealNVP (Dinh et al., 2017) is fast and simple flow architecture but performs poorly compared to more modern methods. Glow (Kingma & Dhariwal, 2018) demonstrates good performance as a generative model but is not state of the art for density estimation, and Residual Flows (Chen et al., 2019) are excellent density estimators but are slower to train than alternatives.

Table 5: Normalizing flow OOD detection vs. architecture comparison. AUROC results are generally comparable, and more performant flow models with superior density estimation do not equate to improved OOD detection performance.

| Backbone Model | Flow Architecture | Out-of-Distribution Dataset (AUROC ↑) | |
| --- | --- | --- | --- |
| | | SVHN | CIFAR-100 |
| | RealNVP | 92.13 | 86.33 |
| ResNet18 + CIFAR-10 | Glow | 96.99 | 85.87 |
| | Residual Flow | 96.89 | 80.32 |

Surprisingly, experiments show that the performance of OOD detection is relatively insensitive to flow architecture. We believe this is due to the fact that discrimination between two distributions (the validation dataset and OOD dataset) is the key task, rather than high quality modeling of the training distribution. As discussed in the previous section, maximum separability (via AUROC) between the validation and OOD datasets is often seen after only a few training epochs of the flow model, far before the training distribution is adequately modelled.

Concisely: the training loss of the flow model is unimportant for OOD detection. Instead, the difference in losses between the ID and OOD distributions is more important. More sophisticated flow models which advance the state of the art in density estimation may not necessarily be advantageous for out-of-distribution detection (see Table 5). In our experiments Glow (Kingma & Dhariwal, 2018) was used as it was found to perform well while being faster than more complex methods, such as Residual flows (Chen et al., 2019) and FFJORD (Grathwohl et al., 2019), and was stable to train.

## 6.5 BACKBONE LATENT REPRESENTATIONS

OOD detection performance is strongly affected by the distribution of latent representations produced by the backbone model in a manner that is independent of image classification accuracy and

difficult to predict. As shown in Section 4, performance varies strongly between convolutional and transformer-based backbones despite similar classification accuracy.

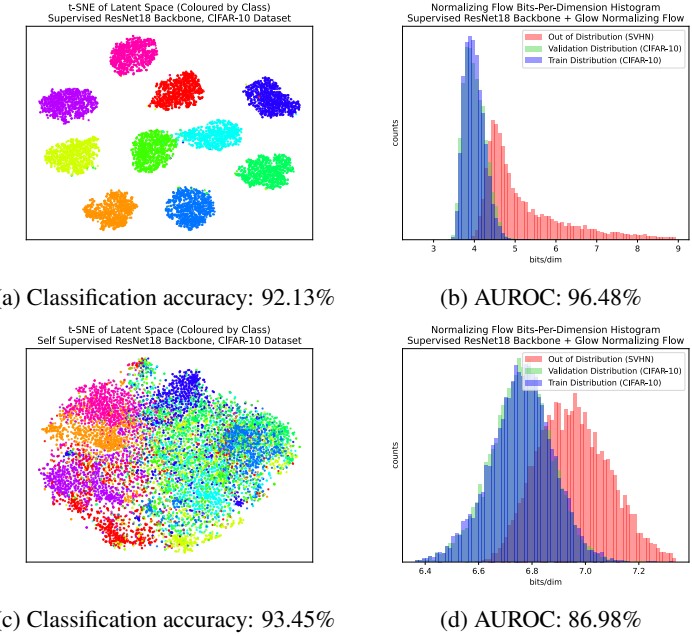

(a) Classification accuracy: 92.13%  (b) AUROC: 96.48%

(c) Classification accuracy: 93.45%  (d) AUROC: 86.98%

Figure 6: Top row: supervised backbone (ResNet18). Bottom row: unsupervised backbone (EMP-SSL ResNet18, Tong et al. (2023)). Note the distinct clusters and good histogram separability for the supervised model, and the more complex latent space with poor histogram separability for the unsupervised model.

Comparison of a supervised and self-supervised ResNet18 CIFAR-10 classifier backbone shows that highly clustered ID latent data (as visualized by t-SNE) is correlated with improved OOD detection performance with this method. We hypothesize that the effectiveness of our proposed method depends on the compactness of latent representations and the margin between in-distribution class boundaries (see Figures 6a, 6c). Specifically, we expect that models with compact ID class representations occupying a lower volume of latent space will increase the likelihood of OOD samples falling in low density regions. Future work will investigate this hypothesis, exploring the connection between pretrained backbone model architecture, ID latent representation compactness, and flow model performance for separating ID and OOD distributions.

## 7 CONCLUSION

We investigate a method for out-of-distribution detection by performing density estimation using normalizing flows in the latent space of pretrained image classification models. In contrast with prior work in this space, our experiments show that by performing density estimation in the latent space and with the identified training regularizations, normalizing flows can achieve strong performance on a variety of common benchmarks on large scale datasets. Our method outperforms all existing methods for detecting far-OOD data, as demonstrated by the results on CIFAR-10 vs. SVHN, and ImageNet-1k vs. Textures.

Performing density estimation on normalized latent variables and under-training the normalizing flow are shown to be particularly important, and we observe the surprising behavior that OOD detection performance peaks very early in training. We further show that OOD detection performance is not dependent on the flow model's ability to perform high quality density estimation, but is strongly dependent on the distribution of latent representations of the backbone model. Using the discussed techniques, we demonstrate that normalizing flows are effective tools for OOD detection in image classification.

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
