# OpenReview forum: "Normalizing Flows For Out of Distribution Detection via Latent Density Estimation"
_ICLR.cc/2024/Conference — Submitted to ICLR 2024_

### Official Review · Reviewer_avuP · 2023-10-18

**Soundness:** 2 fair
**Presentation:** 1 poor
**Contribution:** 2 fair
**Rating:** 3
**Confidence:** 5

**Summary:**

The authors propose applying the latent density estimation via normalizing flows at the last layer of pre-trained classifiers. The method trains a lightweight auxiliary normalizing flow model to perform the out-of-distribution detection via density thresholding. Experimental results are good on incomplete CIFAR10 and ImageNet-1k benchmarks.

**Strengths:**

The only strength to me is the idea of combining generative OOD detection methods with discriminative ones. The idea is novel and interesting--somehow merges the two research lines and proposes a combined method.

**Weaknesses:**

1. **More baselines are needed for comparison.** This method does not belong to either generative or discriminative OOD detection. It uses the activation of the discriminative model to perform generative modeling to get the density. It is somewhere between the two branches of research lines. Thus, I think the model needs to be compared to many other discriminative OOD detection baselines [1,2]. Also, some recent generative OOD detection methods [3,4] need to be compared. At least the authors should discuss the recent approaches in the literature review.

>[1] RankFeat: Rank-1 Feature Removal for Out-of-distribution Detection. NeurIPS 2022.
>
>[2] Extremely Simple Activation Shaping for Out-of-Distribution Detection. ICLR 2023.
>
>[3] The Tiled Variational Autoencoders: Improving Out-of-distribution Detection. ICLR 2023.
>
>[4] Harnessing Out-of-distribution Examples via Augmenting Content and Styles. ICLR 2023.


2. **Missing Details of the CNF of ResNet activations.** The method section is quite unclear to me. The author simply wrote "_learning an invertible mapping between the latent space and a Gaussian probability distribution._" What is the mean and variance of the target Gaussian distribution? How is the model trained? Did you freeze the weight of the main classification branch? Did you still enforce the cross-entropy between class predictions and the ground truth? How do you match the transformed distributions to Gaussian marginals? There are too many important implementation details missing.  After all, it is quite weird that the core method part is quite short and unclear -- even shorter than the introduction of NF and BPD measures.

3. **More OOD datasets are needed for the CIFAR10 benchmark.** Conventionally, OOD detection papers evaluated on CIFAR10 take 6 or 5 OOD datasets, including LSUN-crop, LSUN-resize, iSUN, Places365, and SVHN. However, this paper only chooses SVHN, which does not really capture diverse OOD scenarios and makes the actual average performance across datasets questionable.

4. **Paper Structure.** The paper is poorly written and not organized well. The authors spend 2 pages on related work but use half a page for the method. Actually, some paragraphs in related work are unnecessary, especially the normalizing flow part. Moreover, some subsection in the discussion needs prior knowledge in the method section, such as Sec. 6.3 and 6.2. You need to first tell the audience you perform normalization in the method, and then it makes more sense to discuss why normalization is necessary.

**Questions:**

I strongly suggest the authors detail the methodology section. Given too many important details missing, I can only give reject in its current version.

---

> ### Author Response · Authors · 2023-11-22
>
> Thank you for your feedback, we appreciate your attention to detail.
>
> > More baselines are needed for comparison.
>
> Thank you for the recommended references. We have updated results to include ODIN as an additional benchmark; please note that additional methods do not outperform the existing best baseline, ReAct, on most evaluations. We have also expanded the related works section to discuss more OOD detection methods, including ASH, as recommended.
>
> > Missing Details of the CNF of ResNet activations.
>
> Our methods section (3.1) has been clarified with additional technical details; we hope that the amendments make it straightforward to understand and reproduce our results. Please let us know if you feel additional detail is warranted.
>
> > More OOD datasets are needed for the CIFAR10 benchmark.
>
> We have expanded the set of OOD datasets used for CIFAR-10 evaluation as advised, including Places365, CelebA, and Gaussian noise. In line with our existing observations, our method is generally competitive but excels for far-OOD data, giving strong results for Places365 and CelebA and exceeding other baseline methods for Gaussian noise.
>
> Thank you again for your detailed feedback, we hope that our amendments clarify and address your concerns.

---

### Official Review · Reviewer_gGnx · 2023-10-30

**Soundness:** 2 fair
**Presentation:** 2 fair
**Contribution:** 2 fair
**Rating:** 3
**Confidence:** 3

**Summary:**

The author proposed a post-hoc OOD detection method by training a lightweight auxiliary normalizing flow model and density thresholding. Specifically, the penultimate layer’s activations are used as latent variables for density estimation. The existing normalizing flows are used for learning an invertible mapping between the latent space and a Gaussian probability distribution.

**Strengths:**

1. Previous works assert that normalizing flows are not effective for OOD detection (Kirichenko et al., 2020; Nalisnick et al., 2019), while the authors demonstrate that normalizing flows could achieve competitive results by 1) performing density estimation in the latent space, 2) normalizing the latent representations, and 3) stopping flow training early.

**Weaknesses:**

1. I did not capture the gist of Section 3.2, what is the take-away for this section and how this is connected to the experiments?
2. The authors compare several discriminative model-based OOD detection methods [Energy, MSP], but omit generative model-based OOD detection methods.   For instance: [Provable Guarantees for Understanding Out-of-distribution Detection, AAAI 2022]. Moreover, more advanced OOD detection methods are omit for comparison, such as [ASH,Extremely Simple Activation Shaping for Out-of-Distribution Detection, ICLR 2023].
3. Contribution is limited. [Why Normalizing Flows Fail to Detect Out-of-Distribution Data, NeurIPS 2020] already claimed that normalizing flow on features are better than normalizing flow on the input image.

**Questions:**

Please see the Weeknesses Section.

---

> ### Author Response · Authors · 2023-11-22
>
> Thank you for your review, we appreciate your time and feedback.
>
> > The authors compare several discriminative model-based OOD detection methods [Energy, MSP], but omit generative model-based OOD detection methods. For instance: [Provable Guarantees for Understanding Out-of-distribution Detection, AAAI 2022]. Moreover, more advanced OOD detection methods are omit for comparison, such as [ASH,Extremely Simple Activation Shaping for Out-of-Distribution Detection, ICLR 2023].
>
> We have expanded our related works section accordingly, and incorporated more generative model-based OOD detection methods (although we only use flows a discriminative manner in our work).
>
> > Contribution is limited. [Why Normalizing Flows Fail to Detect Out-of-Distribution Data, NeurIPS 2020] already claimed that normalizing flow on features are better than normalizing flow on the input image.
>
> Thank you for highlighting Kirichenko et al.’s contribution. They do briefly discuss density estimation in the latent space (Sec 8), but their analysis is extremely limited and their results are not strong (Table 2), underperforming other benchmarks we compare to (for example, they report 73.31% AUROC for CIFAR-10 vs SVHN, which all of our tested baselines exceed by at least 13.8%). Our work carries this investigation much further. We have amended our submission to better credit the important contribution of Kirichenko et al.’s prior work, while clarifying our contributions. Our findings that flow models can provide strong post-hoc OOD detection performance on large datasets with latent normalization and undertraining is novel, and we hope these techniques will be a valuable tool for the community.

---

> ### Comment · Reviewer_gGnx · 2023-11-23
>
> I have read the responses and will keep my evaluation unchanged.

---

### Official Review · Reviewer_wdco · 2023-10-31

**Soundness:** 2 fair
**Presentation:** 1 poor
**Contribution:** 1 poor
**Rating:** 3
**Confidence:** 4

**Summary:**

This paper proposes to do density estimation in the learned latent space of normalizing flows for the OOD task in a fully unsupervised way.

**Strengths:**

This method is a post-hoc method that could be applied to any pretrained normalization flows.

**Weaknesses:**

**W1:** The idea is nothing new. Exploiting the latent space of deep generative models has many existing works, but this paper lacks a discussion and comparison of them [1, 2, 3, 4, 5].
Besides, the latent variable's dimension of a flow model is the same as the input image but VAEs latent space dimension could be lower, if the authors claim the property "lightweight", maybe VAEs would be better.

[1] Density of States Estimation for Out of Distribution Detection. AISTATS 2021.

[2] Detecting Out-of-Distribution Inputs to Deep Generative Models Using a Test for Typicality.

[3] Hierarchical VAEs Know What They Don't Know. ICML 2021.

[4] Out-of-distribution detection with an adaptive likelihood ratio on informative hierarchical vae. NeurIPS 2022.

[5] The Tilted Variational Autoencoder: Improving Out-of-Distribution Detection. ICLR 2023.

**W2:** The experiments are not sufficient and convincing, where the compared baselines are limited and too old.

**Questions:**

See the weakness.

---

> ### Author Response · Authors · 2023-11-22
>
> Thank you for highlighting the active field of VAEs for OOD detection. We have expanded our related work section accordingly. We appreciate that you highlighted a key strength of our paper: it is a post-hoc method and is intended to be applied to pretrained classifier models.
>
> > Exploiting the latent space of deep generative models has many existing works, but this paper lacks a discussion and comparison of them [1, 2, 3, 4, 5].
>
> We note that our explanation of our core contributions was perhaps misunderstood. We investigate density estimation in the latent space of a pretrained classifier model, not the latent space of a generative VAE or flow model. The literature listed does not overlap with our main approach: they do not touch on density estimation in the latent spaces of pretrained classifier models, and are not post-hoc methods, but require training of a VAE or flow on pixel data. Because of this training requirement, it may not be trivial to extend the VAE approaches to larger, more complex datasets like full resolution ImageNet. We believe working with pretrained classifier latents remains an important advantage of our method.
>
> Our findings that flow models can provide strong post-hoc OOD detection performance on large datasets with latent normalization and undertraining is novel, and we hope these techniques will be a valuable tool for the community.

---

### Official Review · Reviewer_qftB · 2023-10-31

**Soundness:** 2 fair
**Presentation:** 2 fair
**Contribution:** 1 poor
**Rating:** 5
**Confidence:** 4

**Summary:**

This submission proposes to address the problem of Out-of-distribution (OOD) detection through the use of density estimation under normalizing flow models.

Pre-existing flow architectures are employed to perform density estimation in the latent space of pre-trained classification models. Samples can then be classified as OOD in standard fashion, i.e. if their (latent) representations are evaluated to be of low likelihood under the flow model (are smaller than a threshold value). Quantitative evaluation of the proposed strategy is reported and involves measuring AUROC under six standard datasets used as OOD test scenarios, in comparison with alternative OOD detection methods. Examples of favourable performance are reported.

**Strengths:**

The problem being addressed here is real, and is important -- principled solutions and progress towards techniques that can reliably perform Out-of-distribution detection will be of high value to the community and additionally likely result in useful practical advances. I encourage the authors to consider thinking about these problems. Preliminary investigations into the nature, properties of the latent space with respect to OOD performance (Sec 6.5) are considered interesting.

**Weaknesses:**

As discussed OOD detection is a worthy topic of study, however the current version of the submission raises several key concerns. Namely questions remain over lack of sufficient novelty and contradictions within the crux of the message, in important parts.

Crucially, previous work has already evidenced that 'flows are much better at OOD detection on image embeddings than on the original image datasets'; see Sec. 8, [a]. Somewhat confusingly, the current submission (also) cites [a] when claiming that 'normalizing flows are not effective for OOD detection in this domain'. Authors may wish to comment on this apparent contradiction.

The submission reports upon an experimental search for better performance down various well-trodden paths (e.g. PCA, early stopping, normalisation), however details of innovation are unclear.

Some terminology issues and sloppy writing may serve to distract the reader. Suggest to tighten the exposition and take more care to keep the readership invested. As example:

* Sec 2.3 states both that:
'Normalizing flows [...] have historically performed very poorly for OOD detection in image classification'
and also:
'Zhang et al. demonstrate strong OOD detection performance using normalizing flows in image classification'

* Highlighting problems such as 'few theoretical guarantees' exist and then proceeding with an empirical study leads to confusion. Suggest that authors are instead more focused, in terms of how they prime their readership, for their contributions.

* The abstract claims to present a method 'avoiding researcher bias in OOD sample selection' yet Sec. 6.2 strongly recommends that practitioners 'evaluate [...] on both the original validation and a representative OOD dataset'. Authors may want to extend the latter point by qualifying the aforementioned bias-related issue.

Finally, experimental work can be considered somewhat insufficient. This could be made stronger by considering wider comparisons with additional recent work. There is a large and growing body of OOD work that the authors may wish to also acknowledge and consider (non-exhaustive eg. [b,c,d]).


Summary:

* Concern over the novelty of the hypothesis, insights and sufficiency of both the technical contributions, experimental investigation.

* Contradictory sentiments and statements make it, on occasion, difficult to follow the logical argument and message. Unfortunately the composition of the paper is in a premature state.


 Minor:

* Some figures (e.g. Figure 1, Figure 2) are not referenced in the main body text.
* Tab. 4: typos exist.

References:

a. Kirichenko et al. "Why normalizing flows fail to detect out-of-distribution data". NeurIPS 2020.

b. Liang et al. "Enhancing The Reliability of Out-of-distribution Image Detection in Neural Networks". ICLR 2018.

c. Zhang et al. "On the Out-of-distribution Generalization of Probabilistic Image Modelling". NeurIPS 2021.

d. Wang et al. "Out-of-distribution Detection with Implicit Outlier Transformation". ICLR 2023.

**Questions:**

It has been noted that, in performing OOD detection, the inductive biases of normalizing flows can cause difficulty for image space learning. One interesting question going forward, might be to look at how such biases interact with the noted differences in latent distributions that arise from supervised, unsupervised backbones models (Sec 6.5).

Since flows tend to learn representations that achieve high likelihood through local pixel correlations in the image space, rather than discovering semantic structure, can analogous observations be drawn about differences in the 'shapes' of (un)supervised latent spaces?

**Details Of Ethics Concerns:**

Ethical issues are not discussed by the authors.

---

> ### Author Response · Authors · 2023-11-22
>
> Thank you for your very detailed review.
> > Namely questions remain over lack of sufficient novelty and contradictions within the crux of the message, in important parts
>
> We have updated our submission to emphasize credit given to prior work (in particular Kirichenko et al. 2020) to better clarify the key aspects of our work which are novel, and cleaned up our writing to clarify apparent contradictions.
>
> > The submission reports upon an experimental search for better performance down various well-trodden paths (e.g. PCA, early stopping, normalization), however details of innovation are unclear.
>
> The use of PCA, early stopping, and normalization have not been investigated in depth in the context of using normalizing flows for OOD. Our findings that latent normalization and flow undertraining substantially improve OOD detection is a novel contribution, and we hope the community benefits from these findings. For example, Jiang et al. 2020 applies PCA to reduce their embeddings to 10D for rare example mining; our results show that an unreduced latent space may improve their performance.
>
> > Sec 2.3 states both that: 'Normalizing flows [...] have historically performed very poorly for OOD detection in image classification' and also: 'Zhang et al. demonstrate strong OOD detection performance using normalizing flows in image classification'
> > The abstract claims to present a method 'avoiding researcher bias in OOD sample selection' yet Sec. 6.2 strongly recommends that practitioners 'evaluate [...] on both the original validation and a representative OOD dataset'.
>
> Thank you for the suggestions to improve the presentation of contributions in our work. We have clarified that Zhang et al. is not a post-hoc method and has strong but limited results. We have additionally clarified that all our results are produced with exactly one epoch of flow training to ensure consistent evaluations across models and datasets. We have removed the discussion of guiding early stopping using a representative OOD dataset, as although this technique can further improve OOD performance for a specific OOD set, this distracts from the ability to operate without such OOD sample guidance.
>
> Regarding additional recent works in OOD detection: we have expanded our related works section, added ODIN as a comparison method, and added additional datasets for comparison against CIFAR-10.
>
> > It has been noted that, in performing OOD detection, the inductive biases of normalizing flows can cause difficulty for image space learning. One interesting question going forward, might be to look at how such biases interact with the noted differences in latent distributions that arise from supervised, unsupervised backbones models (Sec 6.5).
> Since flows tend to learn representations that achieve high likelihood through local pixel correlations in the image space, rather than discovering semantic structure, can analogous observations be drawn about differences in the 'shapes' of (un)supervised latent spaces?
>
> We appreciate your curiosity into the results of section 6.5: the connection between the structure and shape of ID data and OOD data in latent space and the performance of our method for OOD detection is also intriguing to us but remains an area of active research. We look forward to expanding on the investigation and results of section 6.5 in future work beyond the scope of this paper.
>
> Thank you for your detailed and focused feedback, we hope our amendments address your concerns.

---

### Official Review · Reviewer_E6xa · 2023-11-01

**Soundness:** 2 fair
**Presentation:** 3 good
**Contribution:** 1 poor
**Rating:** 3
**Confidence:** 4

**Summary:**

In this paper the authors propose using a normalizing flow to model the pre-logit activations of a classification network to detect OOD inputs. They show that flow models can yield good OOD detection performance if they are learned on the normalized latent representations from a pre-trained model backbone, with early stopping.

**Strengths:**

- The paper is written clearly, and I appreciated the in-depth discussion and motivation of the problem of out-of-distribution detection.
- The goal of further investigating the capabilities of normalizing flow models in OOD detection is an important research direction.
- The discussion of key considerations required to achieve good OOD detection performance with normalizing flows shares interesting insights.
- Results on Far-OOD detection are strong

**Weaknesses:**

- The core method of modeling flows in the latent space lacks novelty: [Kirichenko et al., 2020], which the authors cite, explicitly mentions that while flows on the input-space don't perform well, latent-space flows perform much better.
- The authors claim a benefit of their method does not require OOD data. However, in the discussion section, the authors state that representative OOD data is useful to determine when to stop flow training. If early stopping is necessary for good flow performance, the authors should detail how they chose when to stop training in the models used to evaluate the results.
- The experimental evaluation is limited:
	- Results are reported without error bars. Error bars can be computed for a metric like AUROC by using a technique like bootstrapping or cross validation.
	- Missing OOD detection baselines such as ODIN [1] or Mahalanobis Distance based detection [2] which outperform the proposed method on the CIFAR10/SVHN task.
	- Many interesting hypotheses proposed in the discussion section lack quantitative empirical evidence. The observation that compactness of in-distribution class representations in the backbone seemed to correlate with better OOD detection performance was interesting, and worthy of a more extensive study. Adding more datapoints beyond the two shown in the paper (trained supervised vs trained unsupervised), for example by comparing against models with different architectures or different degrees of training performance would significantly strengthen the paper.

**Questions:**

The discussion section of the paper is frank about the practical challenges in getting flow models to work well for OOD detection, and that, in particular, there are considerations on early stopping during training. How did the authors choose when to stop training for the flow models whose OOD detection results are shown in Tables 1 and 2?

---

> ### Author Response · Authors · 2023-11-22
>
> Thank you for your detailed review, we appreciate your feedback and are glad you enjoyed our in-depth discussion.
> > The core method of modeling flows in the latent space lacks novelty: [Kirichenko et al., 2020], which the authors cite, explicitly mentions that while flows on the input-space don't perform well, latent-space flows perform much better.
>
> Thank you for highlighting this. Kirichenko et al. 2020 does briefly discuss density estimation in the latent space (Sec 8), but their analysis is extremely limited and their results do not exceed other state-of-the-art methods in terms of accuracy. Our work carries this investigation much further and demonstrates compelling results that do present NF-based OOD detection as a competitive method. We have amended our work to clarify and credit their prior discussion of this point.
>
> > The authors claim a benefit of their method does not require OOD data. However, in the discussion section, the authors state that representative OOD data is useful to determine when to stop flow training. If early stopping is necessary for good flow performance, the authors should detail how they chose when to stop training in the models used to evaluate the results.
>
> We acknowledge that referring to this as early-stopping was misleading. We have updated our wording to clarify the fixed short-duration training approach and have removed reference to using representative OOD data to determine when to stop flow training. All of our results were produced by training the normalizing flow for exactly one epoch as a matter of convention, as this was found to produce consistently strong results across multiple backbone models and datasets.
>
> > The experimental evaluation is limited:
>
> We have updated results to include ODIN, although additional methods will not outperform the existing best baseline, ReAct, on most evaluations. We have also added additional datasets for evaluation against CIFAR-10. Reporting error bars on AUROC is possible but not widespread in the literature, as output variation is both limited and fairly consistent across methods. We are actively researching the connection between OOD detection and compactness of latent representations (as discussed in sec 6.5) but further discussion is beyond the scope of this paper.
>
> > *Questions:* The discussion section of the paper is frank about the practical challenges in getting flow models to work well for OOD detection, and that, in particular, there are considerations on early stopping during training. How did the authors choose when to stop training for the flow models whose OOD detection results are shown in Tables 1 and 2?
>
> As highlighted in section 6.2, peak OOD detection performance is a function of flow training time that is dependent on the dataset and backbone model. To ensure our method is generally applicable and to avoid reliance on OOD data, all of our results are generated by training the flow model for only one epoch on the training dataset. This produces consistently strong, although not always optimal, results across several backbone models and datasets.

---

### Comment · Area_Chair_1RR8 · 2023-11-22

Dear all,

The author-reviewer discussion period is about to end.

@authors: If not done already, please respond to the comments or questions reviewers may further have. Remain short and to the point.

@reviewers: Please read the author's responses and ask any further questions you may have. To facilitate the decision by the end of the process, please also acknowledge that you have read the responses and indicate whether you want to update your evaluation.

You can update your evaluation positively (if you are satisfied with the responses) or negatively (if you are not satisfied with the responses or share other reviewers' concerns). Please note that major changes are a reason for rejection.

You can also keep your evaluation unchanged. In this case, please indicate that you have read the responses, that you do not have any further comments and that you keep your evaluation unchanged.

Best regards,
The AC

---

### Meta-Review · Area_Chair_1RR8 · 2023-12-08

**Metareview:**

All five reviewers recommend rejection (3-5-3-3-3). The main concerns raised by several reviewers are about the originality of the method, pointing out that Kirichenko et al (2020) have already proposed a similar approach. The author-reviewer discussion has been constructive and has led to a number of improvements to the paper, including a better presentation of previous work and a more thorough experimental evaluation. Despite these improvements, the reviewer decisions remain unchanged.

**Justification For Why Not Higher Score:**

The reviewers are unanimous.

**Justification For Why Not Lower Score:**

N/A

---

### Decision · Program_Chairs · 2024-01-16

Reject